# CONDITIONAL DIFFUSION MODEL FOR WEATHER PREDICTION WITH UNCERTAINTY QUANTIFICATION

## ABSTRACT

Accurate weather forecasting is critical for science and society. Yet, existing methods have not demonstrated high accuracy, low uncertainty, and high computational efficiency simultaneously. On one hand, to quantify the uncertainty in weather predictions, the strategy of ensemble forecast (i.e., generating a set of diverse predictions) is often employed. However, traditional ensemble numerical weather prediction (NWP) is computationally intensive. On the other hand, even though most existing machine learning-based weather prediction (MLWP) approaches are efficient and accurate, they are deterministic and cannot capture the uncertainty of weather forecasting. To tackle these challenges, we propose `CoDiCast`, a conditional diffusion model to generate accurate global weather prediction, while achieving uncertainty quantification and modest computational cost. The key idea behind the prediction task is to generate realistic weather scenarios at a *future* time point, conditioned on observations from the *recent past*. Due to the probabilistic nature of diffusion models, they can be properly applied to capture the uncertainty of weather predictions. Therefore, we accomplish uncertainty quantifications by repeatedly sampling from stochastic Gaussian noise for each initial weather state and running the denoising process multiple times. Experimental results demonstrate that `CoDiCast` outperforms several existing MLWP methods in accuracy, and is faster than NWP models in the inference speed. `CoDiCast` can generate 3-day global weather forecasts, at 6-hour steps and $5.625°$ latitude-longitude resolutions, for over 5 variables, in about 12 minutes on a commodity A100 GPU machine with 80GB memory. The anonymous code is provided at `https://anonymous.4open.science/r/CoDiCast/`.

## 1 INTRODUCTION

Weather prediction describes how the weather states evolve by mapping the current weather states to future weather states (Palmer, 2012). Accurate weather forecasting is crucial for a wide range of societal activities, from daily planning to disaster preparedness (Merz et al., 2020; Shi et al., 2024). For example, governments, organizations, and individuals rely heavily on weather forecasts to make informed decisions that can significantly impact safety, economic efficiency, and overall well-being. However, weather predictions are intrinsically uncertain largely due to the complex and chaotic nature of atmospheric processes (Slingo & Palmer, 2011; Palmer et al., 2005). Therefore, assessing the range of probable weather scenarios is significant, as it facilitates informed decision-making.

Traditional numerical weather prediction (NWP) methods achieve weather forecasting by approximately solving the differential equations representing the integrated system between the atmosphere, land, and ocean (Price et al., May 2024; Nguyen et al., 2023). However, running such an NWP model can produce only one possibility of the forecast, which ignores the weather uncertainty. To solve this problem, *Ensemble forecast*[1] of multiple models is often employed to model the probability distribution of different future weather scenarios (Palmer, 2019; Leinonen et al., 2023). While such NWP-based ensemble forecasts effectively model the weather uncertainty, they have two primary limitations: physics-based models inherently make restrictive assumptions of atmospheric dynamics (Palmer et al., 2005) and running multiple these NWP-models require extreme computational costs (Rodwell & Palmer, 2007).

---

[1]Generating a set of forecasts, each of which represents a single possible scenario.

In recent years, machine learning (ML)-based weather predictions (MLWP) have been proposed to challenge NWP-based prediction methods (Ben Bouallègue et al., 2024; Bülte et al., 2024). They have achieved enormous success with comparable accuracy and a much (usually three orders of magnitude) lower computational overhead. Representative work includes Pangu (Bi et al., 2023), GraphCast (Lam et al., 2023), ClimaX (Nguyen et al., 2023), ForeCastNet (Pathak et al., 2022), Fuxi (Chen et al., 2023b), Fengwu (Chen et al., 2023a), and W-MAE (Man et al., 2023). They are typically trained to learn weather patterns from a huge amount of historical data and predict the mean of the probable trajectories by minimizing the mean squared error (MSE) of model forecasts (Hewage et al., 2021). Despite the notable achievements of these MLWP methods, most of them are deterministic (Kochkov et al., 2024), falling short in capturing the uncertainty in weather forecasts (Jaseena & Kovoor, 2022). This limitation motivates us to explore an approach for uncertainty quantification while being capable of forecasting weather scenarios accurately.

Denoising probabilistic diffusion models (DDPMs) (Ho et al., 2020) stand out as a probabilistic type of generative models, which can generate high-quality image samples. By explicitly and iteratively modeling the noise additive and its removal, DDPMs can capture intricate details and textures of images. Furthermore, controllable diffusion models (Rombach et al., 2022; Zhang et al., 2023) enable the generation process to be guided by specific attributes or conditions, e.g., class labels, textual descriptions, or other auxiliary information. By doing so, the models can generate images that adhere to the specified conditions. This inspires us to consider the weather "prediction" tasks as "generation" tasks - generating plausible weather scenarios with conditional diffusion models. Promising potentials could be the following: (1) Weather numerical data is usually a 2-D grid over latitude and longitude, sharing a similar modality with the image. Diffusion models can capture the intricate weather distribution with iterative denoising. (2) Weather states from the recent past (i.e., initial conditions) can be injected into diffusion models to guide the generation of future weather evolution. (3) More notably, probabilistic diffusion models can generate a set of diverse weather scenarios rather than a single deterministic one. This capability makes them well-suited for modeling the uncertain nature of weather evolution. Our contributions are presented as follows:

- We identify the shortcomings of current weather prediction methods. NWP-based methods are limited to restrictive assumptions and computationally intensive. Moreover, a single deterministic NWP- and MLWP-based method cannot achieve uncertainty quantification.

- To address these problems, we propose `CoDiCast`, a conditional diffusion model for global weather prediction conditioning on observations from the recent past while probabilistically modeling the uncertainty. In addition, we use the cross-attention mechanism to effectively integrate conditions into the denoising process to guide the generation tasks.

- We conduct extensive experiments on a decade of ERA5 reanalysis data from the European Centre for Medium-Range Weather Forecasts (ECMWF), and evaluate our method against several state-of-the-art models in terms of accuracy, efficiency, and uncertainty. It turns out that `CoDiCast` achieves an essential trade-off among these valuable properties.

## 2 RELATED WORK

**Numerical Weather Prediction.** Numerical Weather Prediction (NWP) methods obtain weather forecasts by modeling the system of the atmosphere, land, and ocean with complex differential equations (Bauer et al., 2015). High-Resolution Forecasts System (HRES) (ECMWF, 2023) is a representative NWP method that forecasts possible weather evolution out to 10 days ahead. However, HRES is a deterministic NWP method that only provides a single forecast. The ensemble forecast suite (ENS) (Buizza, 2008) was developed as an ensemble of 51 forecasts by the European Centre for Medium-Range Weather Forecasts (ECMWF). ENS provides a range of possible future weather states in the medium range, allowing for investigation of the detail and uncertainty in the forecast. Even if ENS and other NWP-based ensemble forecasts effectively model the weather evolution, they exhibit sensitivity to structural discrepancies across models (Balaji et al., 2022), regional variability (Verma et al., 2024), and high computational demands (Lam et al., 2023).

**ML-Based Weather Prediction.** Numerous machine learning (ML)-based weather prediction (MLWP) approaches have emerged as a compelling alternative to NWP methods on weather forecasting. They are trained on enormous historical data and produce the mean of the probable

trajectories by minimizing the mean squared error (MSE) between model forecasts and ground-truth (Hewage et al., 2021). Pangu (Bi et al., 2023) employed three-dimensional transformer networks and Earth-specific priors to deal with complex patterns in weather data. GraphCast (Lam et al., 2023) achieved medium-range weather prediction by utilizing an "encode-process-decode" configuration with each part implemented by graph neural networks (GNNs). GNNs perform effectively in capturing the complex relationship between a set of surface and atmospheric variables. A similar GNN-based work is (Keisler, 2022). Fuxi (Chen et al., 2023b) and Fengwu (Chen et al., 2023a) also employ the "encode-decode" strategy but with the transformer-based backbone. FourCastNet (Pathak et al., 2022) applied Vision Transformer (ViT) (Dosovitskiy et al., 2020) and Adaptive Fourier Neural Operators (AFNO) (Guibas et al., 2021), while ClimaX (Nguyen et al., 2023) also uses a ViT backbone but the trained model can be fine-tuned to various downstream tasks. However, these models fall short in modeling the uncertainty of weather evolution (Jaseena & Kovoor, 2022; Bülte et al., 2024). Additionally, ClimODE (Verma et al., 2024) incorporated the physical knowledge and developed a continuous-time neural advection PDE weather model.

**Diffusion Models.** Diffusion models (Ho et al., 2020; Rombach et al., 2022) have shown their strong capability in computer vision tasks, including image generation (Li et al., 2022), image editing (Nichol et al., 2021), semantic segmentation (Brempong et al., 2022) and point cloud completion (Luo & Hu, 2021). Conditional diffusion models (Ho & Salimans, 2022) were later proposed to make the generation step conditioned on the current context or situation. However, not many efforts have adopted diffusion models in global medium-range weather forecasting. More recent research has focused on precipitation nowcasting (Asperti et al., 2023; Gao et al., 2024; Yu et al., 2024), and are localized in their predictions. GenCast (Price et al., May 2024) is a recently proposed close-sourced conditional diffusion-based ensemble forecasting for medium-range weather prediction. However, the conditioning is shown to be insufficient in our paper (see the last case in ablation study). Since GenCast is not open-sourced, we do not have access to details for a fair comparison.

## 3 PRELIMINARIES

In this section, we introduce the problem formulation of global weather prediction and briefly review Denoising Diffusion Probabilistic Models (DDPMs) (Ho et al., 2020).

### 3.1 PROBLEM FORMULATION

**Deterministic Global Weather Predictions.** Given the input consisting of the weather state(s), $X^t \in \mathbb{R}^{H \times W \times C}$ at time $t$, the problem is to predict a point-valued weather state, $X^{t+\Delta t} \in \mathbb{R}^{H \times W \times C}$ at a future time point $t + \Delta t$. $H \times W$ refers to the spatial resolution of data which depends on how densely we grid the globe over latitudes and longitudes, $C$ refers to the number of channels (i.e., weather variables), and the superscripts $t$ and $t + \Delta t$ refer to the current and future time points. The long-range multiple-step forecasts could be achieved by autoregressive modeling or direct predictions.

**Probabilistic Global Weather Predictions.** Unlike the deterministic models that output point-valued predictions, probabilistic methods model the probability of future weather state(s) as a distribution $P(X^{t+\Delta t} \mid X^t)$, conditioned on the state(s) from the recent past. Probabilistic predictions are appropriate for quantifying the forecast uncertainty and making informed decisions.

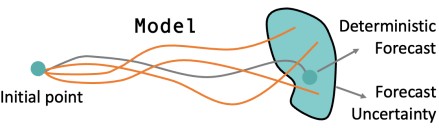

Figure 1: Deterministic vs Probabilistic.

### 3.2 DENOISING DIFFUSION PROBABILISTIC MODELS

Denoising diffusion probabilistic model (DDPM) (Ho et al., 2020) generates target samples by learning a distribution $p_\theta(x_0)$ that approximates the target distribution $q(x_0)$. DDPM comprises a *forward diffusion* process and a *reverse denoising* process.

The *forward process* involves no learnable parameters and transforms an input $x_0$ with a data distribution of $q(x_0)$ to a white Gaussian noise vector $x_N$ in $N$ diffusion steps. It can be described

as a Markov chain that gradually adds Gaussian noise to the input according to a variance schedule $\{\beta_1, \ldots, \beta_N\}$:

$$q(x_{1:N} \mid x_0) = \prod_{n=1}^{N} q(x_n \mid x_{n-1}), \tag{1}$$

where at each step $n \in [1, N]$, the diffused sample $x_n$ is obtained $q(x_n \mid x_{n-1}) = \mathcal{N}\left(x_n; \sqrt{1-\beta_n}x_{n-1}, \beta_n \mathbf{I}\right)$. Instead of iteratively sampling $x_n$ step by step following the chain, the forward process enables sampling $x_n$ at an arbitrary step $n$ in the closed form:

$$q(x_n \mid x_0) = \mathcal{N}\left(x_n; \sqrt{\bar{\alpha}_n}x_0, (1-\bar{\alpha}_n)\mathbf{I}\right), \tag{2}$$

where $\alpha_n = 1 - \beta_n$ and $\bar{\alpha}_n = \prod_{s=1}^{n} \alpha_s$. Thus, $x_n$ can be directly obtained as $x_n = \sqrt{\bar{\alpha}_n}x_0 + \sqrt{1-\bar{\alpha}_n}\epsilon$ with $\epsilon$ is sampled from $\mathcal{N}(\mathbf{0}, \mathbf{I})$.

In the *reverse process*, the *denoiser* network is used to recover $x_0$ by gradually denoising $x_n$ starting from a Gaussian noise $x_N$ sampled from $\mathcal{N}(\mathbf{0}, \mathbf{I})$. This process is formally defined as:

$$p_\theta(x_{0:N}) = p(x_N) \prod_{n=1}^{N} p_\theta(x_{n-1} \mid x_n), \tag{3}$$

where the data distributions, parameterized by $\theta$, are represented as $p_\theta(x_n), p_\theta(x_{n-1}), \ldots, p_\theta(x_0)$.

For each diffusion iteration $n \in \{1, 2, \ldots, N\}$, diffusion models can be trained to minimize the following KL-divergence:

$$\mathcal{L}_n = D_{KL}\left(q(x_{n-1} \mid x_n) \parallel p_\theta(x_{n-1} \mid x_n)\right). \tag{4}$$

where $q(x_{n-1} \mid x_n)$ is often replaced by:

$$q(x_{n-1} \mid x_n, x_0) = \mathcal{N}\left(x_{n-1}; \tilde{\mu}_n(x_n, x_0, n), \tilde{\beta}_n\right), \tag{5}$$

and $p_\theta(x_{n-1} \mid x_n)$ is represented by:

$$p_\theta(x_{n-1} \mid x_n) = \mathcal{N}\left(x_{n-1}; \mu_\theta(x_n, n), \Sigma_\theta(x_n, n)\right). \tag{6}$$

In practice, $\Sigma_\theta(x_n, n)$ is fixed at $\tilde{\sigma}_n^2 \mathbf{I}$ where $\tilde{\sigma}_n^2 = \tilde{\beta}_n = \beta_n \frac{1-\bar{\alpha}_{k-1}}{1-\bar{\alpha}_k}$, and $\mu_\theta(x_n, n)$ is modeled by denoiser, a neural network parameterized by $\theta$. Therefore, comparing Eq. (6) and Eq. (5), the loss function in Eq. (4) is transformed to:

$$\mathcal{L}_n = \frac{1}{2\tilde{\sigma}_n^2} \|\tilde{\mu}_n(x_n, x_0, n) - \mu_\theta(x_n, n)\|^2, \tag{7}$$

where

$$\tilde{\mu}(x_n, x_0, n) = \frac{1}{\sqrt{\alpha_n}}\left(x_n - \frac{1-\alpha_n}{\sqrt{1-\bar{\alpha}_n}}\epsilon_n\right), \tag{8}$$

$$\mu_\theta(x_n, n) = \frac{1}{\sqrt{\alpha_n}}\left(x_n - \frac{1-\alpha_n}{\sqrt{1-\bar{\alpha}_n}}\epsilon_\theta(x_n, n)\right). \tag{9}$$

Now, the loss function above can be simplified to Eq. (10). Each diffusion step $n$ simply minimizes the difference between the noise added in the forward process and the one from the denoiser output. DDPM (Ho et al., 2020) claims that such a simplified loss function is easy to train and beneficial for generating samples of better quality.

$$\mathcal{L}_{simple}(\theta) = \mathbb{E}_{x_0, \epsilon, n} \|\epsilon - \epsilon_\theta(x_n, n)\|^2, \tag{10}$$

where $\epsilon_\theta(\cdot)$ is a denoiser network to predict the added noise in the forward process. Once trained, target variables are first sampled from Gaussian as the input of $\epsilon_\theta(\cdot)$ to progressively learn the distribution $p_\theta(x_{n-1}|x_n)$ and denoise $x_n$ until $x_0$ is obtained, as shown in Eq. (3).

## 4 METHODOLOGY

This section introduces our approach for global weather prediction, CoDiCast, implemented as a conditional diffusion model. The key idea is to consider "prediction" tasks as "generation" tasks while conditioning on the context guidance of past observation(s). An overview of the proposed CoDiCast is shown in Figure 2.

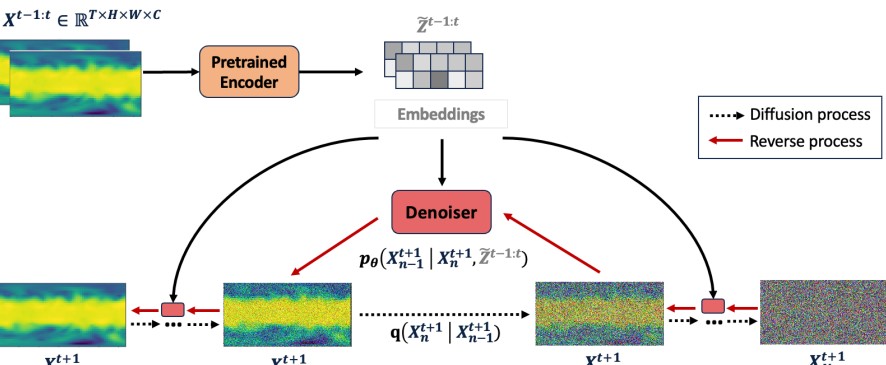

Figure 2: Framework of `CoDiCast` for global weather forecast. The superscript $T$ and the subscript $N$ denote the time point and iteration step of adding/denoising noise. $H$ and $W$ represent the height (#latitude) and width (#longitude) of grid data. $C$ is the number of variables of interest.

## 4.1 FORWARD DIFFUSION PROCESS

The forward diffusion process is straightforward. Assuming the current time point is $t$, for the sample at time point $t + 1$, $X_0^{t+1} \in \mathbb{R}^{H \times W \times C}$, which is of interest to predict, we first compute the diffused sample by gradually adding noise until the $N^{th}$ iteration (see the dotted lines in Figure 2):

$$X_n^{t+1} = \sqrt{\bar{\alpha}_n} \cdot X_0^{t+1} + \sqrt{1 - \bar{\alpha}_n}\epsilon, \tag{11}$$

where $\epsilon$ is sampled from $\mathcal{N}(\mathbf{0}, \mathbf{I})$ with the same size as $X_0^{t+1}$, and $\bar{\alpha}$ is same as that in Eq. (2).

## 4.2 REVERSE CONDITIONAL DENOISING PROCESS

`CoDiCast` models the probability distribution of the future weather state conditioning on the current and previous weather states. More specifically, we exploit a pre-trained encoder to learn conditions as embedding representations of the past observations $X^{t-1}$ and $X^t$, which are used to control and guide the synthesis process. Compared to modeling the past observations in the original space, we found that our embedding representations in the latent space work better.

$$p_\theta(X_{0:N}^{t+1} \mid \tilde{Z}^{t-1:t}) = p(X_N^{t+1}) \prod_{n=1}^{N} p_\theta(X_{n-1}^{t+1} \mid X_n^{t+1}, \tilde{Z}^{t-1:t}), \tag{12}$$

where $X_N^{t+1} \sim \mathcal{N}(\mathbf{0}, \mathbf{I})$, $\tilde{Z}^{t-1:t}$ is the embedding representation as shown in Eq. (14).

After prediction at the first time point is obtained, a forecast trajectory, $X^{1:T}$, of length $T$, can be auto-regressively modeled by conditioning on the predicted "previous" states.

$$p_\theta(X_{0:N}^{1:T}) = \prod_{t=1}^{T} p(X_N^t) \prod_{n=1}^{N} p_\theta(X_{n-1}^t \mid X_n^t, \tilde{Z}^{t-2:t-1}). \tag{13}$$

## 4.3 PRE-TRAINED ENCODER

We learn an encoder by training an autoencoder network (Baldi, 2012). An `Encoder` compresses the input at each time point into a latent-space representation, while `Decoder` reconstructs the input from the latent representation. After the encoder, $\mathcal{F}$, is trained, it can serve as a pre-trained representation learning model to project the original data into latent embedding in Eq. (14). Appendix B.1 provides more details.

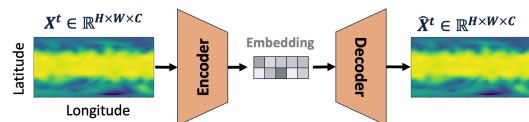

Figure 3: Autoencoder structure.

$$\tilde{Z}^{t-1:t} = \mathcal{F}(X^{t-1}, X^t) \tag{14}$$

## 4.4 ATTENTION-BASED DENOISER NETWORK

Figure 4: Attention-based denoiser structure.

Our denoiser network consists of two blocks: cross-attention and U-net (as shown in Figure 4). Cross-attention mechanism (Hertz et al., 2022) is employed to capture how past observations can contribute to the generation of future states. The embedding of past observations, $\tilde{Z}^{t-1:t}$, and the noise data $X_n^{t+1}$ at diffusion step $n$, are projected to the same hidden dimension $d$ with the following transformation:

$$Q = W_q \cdot X_n^{t+1}, K = W_k \cdot \tilde{Z}^{t-1:t}, V = W_v \cdot \tilde{Z}^{t-1:t}, \tag{15}$$

where $X_n^{t+1} \in \mathbb{R}^{(H \times W) \times C}$ and $\tilde{Z}^{t-1:t} \in \mathbb{R}^{(H \times W) \times d_z}$. $W_q \in \mathbb{R}^{d \times C}, W_k \in \mathbb{R}^{d \times d_z}, W_v \in \mathbb{R}^{d \times d_z}$ are learnable projection matrices (Vaswani et al., 2017). Then we implement the cross-attention mechanism by `Attention(Q, K, V)` = `softmax`$(\frac{QK^T}{\sqrt{d}})V$. A visual depiction of the cross-attention mechanism is in Appendix B.2.

U-Net (Ronneberger et al., 2015) is utilized to recover the data by removing the noise added at each diffusion step. The *skip connection* technique in U-Net concatenates feature maps from the encoder to the corresponding decoder layers, allowing the network to retain fine-grained information that might be lost during downsampling. The detailed U-Net architecture is presented in Appendix B.3.

## 4.5 TRAINING PROCESS

The training procedure is shown in Algorithm 1. Firstly, we pre-train an `encoder` to learn the condition embedding of the past observations. Subsequently, we inject it into our conditional diffusion model and train `CoDiCast` with the devised loss function:

$$\mathcal{L}_{cond}(\theta) = \mathbb{E}_{X_0, \epsilon, n} \left\| \epsilon - \epsilon_\theta \left( X_n^{t+1}, n, \texttt{cond} \right) \right\|^2, \tag{16}$$

where $X_n^{t+1} = \sqrt{\bar{\alpha}_n} X_0^{t+1} + \sqrt{1 - \bar{\alpha}_n}\epsilon$, `cond` $= \mathcal{F}(X^{t-1:t})$, and $\epsilon_\theta$ is the denoiser in Figure 4.

## 4.6 INFERENCE PROCESS

Algorithm 2 describes the inference process. We first extract the conditional embedding representations, $\tilde{Z}^{t-1:t}$, by the pre-trained encoder, and then randomly generate a noise vector $X_N \sim \mathcal{N}(\mathbf{0}, \mathbf{I})$ of size $H \times W \times C$. The sampled noise vector, $X_N$, is autoregressively denoised along the reversed chain to predict the target until $n$ equals 1 ($\zeta$ is set to zero when $n = 1$), we obtain weather prediction $\hat{X}_0$ at the time $t + 1$. Later, multi-step prediction can be implemented autoregressively - the output from the previous time step is the input while predicting the next step, as shown in Eq. (13).

---

**Algorithm 1** Training

1: **Input**: Number of diffusion steps $N$, pre-trained encoder $\mathcal{F}$
2: **Output**: Trained denoising function $\epsilon(\cdot)$
3: **repeat**
4:     $X_0^{t+1} \sim q(X_0^{t+1})$
5:     $n \sim \texttt{Uniform}(1, 2, \ldots, N)$
6:     $\epsilon \sim \mathcal{N}(\mathbf{0}, \mathbf{I})$
7:     Get the past observations $X^{t-1}, X^t$
8:     Get embedding $\tilde{Z}^{t-1:t} = \mathcal{F}(X^{t-1}, X^t)$
9:     Take gradient descent step on:

$$\nabla_\theta \left\| \epsilon - \epsilon_\theta \left( X_n^{t+1}, n, \tilde{Z}^{t-1:t} \right) \right\|^2$$

10: **until** converged

---

**Algorithm 2** Inference

1: **Input**: Number of diffusion steps $N$, pre-trained encoder $\mathcal{F}$, trained denoising network $\epsilon(\cdot)$, past observations $X^{t-1}, X^t$
2: **Output**: Inference target $X_0^{t+1}$
3: Get embedding $\tilde{Z}^{t-1:t} = \mathcal{F}(X^{t-1}, X^t)$
4: $X_N \sim \mathcal{N}(\mathbf{0}, \mathbf{I})$
5: **for** $n = N, \ldots, 1$ **do**
6:     $\zeta \sim \mathcal{N}(\mathbf{0}, \mathbf{I})$ if $n \geq 1$, else $\zeta = 0$
7:     $X_{n-1}^{t+1} = \frac{1}{\sqrt{\alpha_n}} \left( X_n^{t+1} - \frac{1 - \alpha_n}{\sqrt{1 - \bar{\alpha}_n}} \epsilon_\theta(X_n^{t+1}, n, \tilde{Z}^{t-1:t}) \right) + \sigma_n \zeta$
8: **end for**
9: **return** $X_0^{t+1}$

---

### 4.7 ENSEMBLE FORECAST

To enhance the reliability of weather forecasts, *ensemble forecast* strategy is often employed to capture the variability among forecasts by separately running multiple deterministic models, e.g., ensemble forecast suite (ENS) (Buizza, 2008). In our approach, since `CoDiCast` is a probabilistic model that can generate a distribution of future weather scenarios rather than a single prediction, following (Price et al., May 2024), we run the trained `CoDiCast` multiple times to get the ensemble instead. More specifically, by integrating both initial conditions and noise sampled from a Gaussian distribution, `CoDiCast` implements the ensemble forecast through multiple stochastic samplings during inference, capturing a range of possible forecasts for the uncertainty quantification.

## 5 EXPERIMENTS

### 5.1 DATASET AND BASELINES

**Dataset.** ERA5 (Hersbach et al., 2020) is a publicly available atmospheric reanalysis dataset provided by the European Centre for Medium-Range Weather Forecasts (ECMWF). Following the existing work (Verma et al., 2024), we use the preprocessed $5.625°$ resolution ($32 \times 64$) and 6-hour increment ERA5 dataset from WeatherBench (Rasp et al., 2020). We downloaded 5 variables for the globe: geopotential at 500 hPa pressure level (`Z500`), atmospheric temperature at 850 hPa pressure level (`T850`), ground temperature (`T2m`), 10 meter U wind component (`U10`) and 10 meter V wind component (`V10`). More details can be found in Table 4 in Appendix A.

**Baselines.** We comprise the following methods as baselines. The first four ML benchmarks use the same data set described in Section 5.1 for a fair comparison. We are unable to compare against Pangu-Weather (Bi et al., 2023) and Graphcast (Lam et al., 2023) due to the various resolutions they used, their partially released code, and our limited computing resources.

- **ClimODE** (Verma et al., 2024): a spatiotemporal continuous-time model that incorporates the physic knowledge of atmospheric *advection* over time.
- **ClimaX** (Nguyen et al., 2023): a state-of-the-art vision Transformer-based method trained on the same dataset (without pre-training that is used in the original paper).
- **FourCastNet (FCN)** (Pathak et al., 2022): a global data-driven weather model using adaptive Fourier neural operators.
- **Neural ODE** (Chen et al., 2018): an ODE network that learns the time derivatives as neural networks by solving an ordinary differential equation.
- **Integrated Forecasting System IFS** (Rasp et al., 2020): one of the most advanced global numerical weather prediction (NWP) models. IFS is often viewed as the gold standard.

### 5.2 EXPERIMENTS DESIGN

We use data between 2006 and 2015 as the training set, data in 2016 as the validation set, and data between 2017 and 2018 as the testing set. We assess the global weather forecasting capabilities of our method `CoDiCast` by predicting the weather at a future time $t + \Delta t$ ($\Delta t$ = 6 to 36 hours) based on the past two time units. To quantify the uncertainty in weather prediction, we generate an "ensemble" forecast by running `CoDiCast` three times during the inference phase. We also analyze and compare the inference efficiency between the NWP-based methods and `CoDiCast`.

**Training.** We first pretrain an `encoder` with the `Autoencoder` architecture. For the diffusion model, we used U-Net as the denoiser network with 1000 diffusion/denoising steps. The architecture is similar to that of DDPM (Ho et al., 2020) work. We employ four U-Net units for both the downsampling and upsampling processes. Each U-Net unit comprises two ResNet blocks (He et al., 2016) and a convolutional up/downsampling block. Before training, we apply Max-Min normalization (Ali et al., 2014) to scale the input data within the range $[0, 1]$, mitigating potential biases stemming from varying scales (Shi et al., 2023). `Adam` was used as the optimizer, where the learning rate $= 2e^{-4}$, decay steps $= 10000$, decay rate $= 0.95$. The batch size and number of epochs were set to 64 and 800 respectively. More training details and model configurations are in Appendix C.

**Evaluation Metrics.** Following (Verma et al., 2024), we use latitude-weighted Root Mean Square Error (RMSE) and Anomaly Correlation Coefficient (ACC) as deterministic metrics. RMSE measures the average difference between values predicted by a model and the actual values. ACC is the correlation between prediction anomalies relative to climatology and ground truth anomalies relative to climatology. It is a critical metric in climate science to evaluate the model's performance in capturing unusual weather or climate events. Moreover, following (Rasp et al., 2024) we utilize the continuous ranked probability score (CRPS) (Gneiting & Raftery, 2007) as a probabilistic metric to measure the discrepancy between the predicted distribution and a single ground-truth value. A lower CRPS value indicates higher forecast accuracy. Appendix D contains the formulas of these metrics.

## 5.3 QUANTITATIVE EVALUATION

**Accuracy.** We compare different models in forecasting five primary meteorological variables as described in Section 5.1. From Table 1, we observe that CoDiCast presents superior performance across latitude-weighted RMSE metrics over other MLWP baselines while it shows comparable performance across ACC scores. In Appendix E, we provide the predictions with longer lead times (up to 6 days). However, CoDiCast still falls short in comparison with the gold-standard IFS model.

Table 1: Latitude-weighted RMSE (↓) and ACC (↑) comparison with baselines on global weather forecasting. We mark the scores in bold if CoDiCast performs the best among MLWP methods.

| Variable | Lead Time | RMSE (↓) | | | | | | ACC (↑) | | | | | |
|---|---|---|---|---|---|---|---|---|---|---|---|---|---|
| | | NODE | ClimaX | FCN | IFS | ClimODE | CoDiCast | NODE | ClimaX | FCN | IFS | ClimODE | CoDiCast |
| Z500 | 6 | 300.6 | 247.5 | 149.4 | 26.9 | 102.9±9.3 | **73.1**±6.7 | 0.96 | 0.97 | 0.99 | 1.00 | 0.99 | **0.99** |
| | 12 | 460.2 | 265.3 | 217.8 | N/A | 134.8±12.3 | **114.2**±8.9 | 0.88 | 0.96 | 0.99 | N/A | 0.99 | **0.99** |
| | 18 | 627.6 | 319.8 | 275.0 | N/A | 162.7±14.4 | **152.4**±10.4 | 0.79 | 0.95 | 0.99 | N/A | 0.98 | **0.99** |
| | 24 | 877.8 | 364.9 | 333.0 | 51.0 | 193.4±16.3 | **186.5**±11.8 | 0.70 | 0.93 | 0.99 | 1.00 | 0.98 | 0.98 |
| | 36 | 1028.2 | 455.0 | 449.0 | N/A | 259.6±22.3 | **256.7**±14.6 | 0.55 | 0.89 | 0.99 | N/A | 0.96 | 0.97 |
| T850 | 6 | 1.82 | 1.64 | 1.18 | 0.69 | 1.16±0.06 | **1.02**±0.05 | 0.94 | 0.94 | 0.99 | 0.99 | 0.97 | **0.99** |
| | 12 | 2.32 | 1.77 | 1.47 | N/A | 1.32±0.13 | **1.26**±0.10 | 0.85 | 0.93 | 0.99 | N/A | 0.96 | **0.99** |
| | 18 | 2.93 | 1.93 | 1.65 | N/A | 1.47±0.16 | **1.41**±0.12 | 0.77 | 0.92 | 0.99 | N/A | 0.96 | 0.97 |
| | 24 | 3.35 | 2.17 | 1.83 | 0.87 | 1.55±0.18 | **1.52**±0.16 | 0.72 | 0.90 | 0.99 | 0.99 | 0.95 | 0.97 |
| | 36 | 4.13 | 2.49 | 2.21 | N/A | 1.75±0.26 | **1.75**±0.19 | 0.58 | 0.86 | 0.99 | N/A | 0.94 | 0.96 |
| T2m | 6 | 2.72 | 2.02 | 1.28 | 0.97 | 1.21±0.09 | **0.95**±0.07 | 0.82 | 0.92 | 0.99 | 0.99 | 0.97 | **0.99** |
| | 12 | 3.16 | 2.26 | 1.48 | N/A | 1.45±0.10 | **1.21**±0.07 | 0.68 | 0.90 | 0.99 | N/A | 0.96 | **0.99** |
| | 18 | 3.45 | 2.45 | 1.61 | N/A | 1.43±0.09 | **1.34**±0.08 | 0.69 | 0.88 | 0.99 | N/A | 0.96 | **0.99** |
| | 24 | 3.86 | 2.37 | 1.68 | 1.02 | 1.40±0.09 | **1.45**±0.07 | 0.79 | 0.89 | 0.99 | 0.99 | 0.96 | 0.98 |
| | 36 | 4.17 | 2.87 | 1.90 | N/A | 1.70±0.15 | **1.65**±0.11 | 0.49 | 0.83 | 0.99 | N/A | 0.94 | 0.97 |
| U10 | 6 | 2.30 | 1.58 | 1.47 | 0.80 | 1.41±0.07 | **1.24**±0.06 | 0.85 | 0.92 | 0.95 | 0.98 | 0.91 | **0.95** |
| | 12 | 3.13 | 1.96 | 1.89 | N/A | 1.81±0.09 | **1.50**±0.08 | 0.70 | 0.88 | 0.93 | N/A | 0.89 | **0.93** |
| | 18 | 3.41 | 2.24 | 2.05 | N/A | 1.97±0.11 | **1.68**±0.08 | 0.58 | 0.84 | 0.91 | N/A | 0.88 | **0.91** |
| | 24 | 4.10 | 2.49 | 2.33 | 1.11 | 2.01±0.10 | **1.87**±0.09 | 0.50 | 0.80 | 0.89 | 0.97 | 0.87 | **0.89** |
| | 36 | 4.68 | 2.98 | 2.87 | N/A | 2.25±0.18 | **2.25**±0.12 | 0.35 | 0.69 | 0.85 | N/A | 0.83 | **0.87** |
| V10 | 6 | 2.58 | 1.60 | 1.54 | 0.94 | 1.53±0.08 | **1.30**±0.06 | 0.81 | 0.92 | 0.94 | 1.00 | 0.92 | **0.95** |
| | 12 | 3.19 | 1.97 | 1.81 | N/A | 1.81±0.12 | **1.56**±0.09 | 0.61 | 0.88 | 0.91 | N/A | 0.89 | **0.93** |
| | 18 | 3.58 | 2.26 | 2.11 | N/A | 1.96±0.16 | **1.75**±0.11 | 0.46 | 0.83 | 0.86 | N/A | 0.88 | **0.91** |
| | 24 | 4.07 | 2.48 | 2.39 | 1.33 | 2.04±0.10 | **1.94**±0.14 | 0.35 | 0.80 | 0.83 | 1.00 | 0.86 | **0.89** |
| | 36 | 4.52 | 2.98 | 2.95 | N/A | 2.29±0.24 | **2.35**±0.18 | 0.29 | 0.69 | 0.75 | N/A | 0.83 | **0.85** |

**Uncertainty.** The error range (in gray) associated with our CoDiCast in Table 1 is smaller than ClimODE, indicating that our method can produce more robust predictions. We provide a case study of CoDiCast forecast for 72 hours with uncertainty quantification in Figure 5. It shows the mean prediction tracks the general trend of the ground truth and the uncertainty grows as the lead time increases. Besides, most actual values fall within the 1 or 2 standard deviation ($\sigma$) ranges, indicating that predictions are reasonably accurate but could be improved for higher precision. We also report CRPS scores with

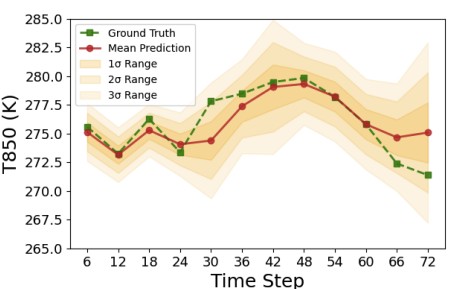

Figure 5: Forecast with confidence intervals.

24-hour prediction in Table 2. Because most existing MLWP methods produce deterministic forecasts, we use IFS ENS, an ensemble of 51 NWP-based forecasts (Buizza, 2008) for a relative reference. We observe that there is still room for CoDiCast to be improved, but we claim it achieves valuable probabilistic forecasts and the inference speed is faster than IFS ENS (next subsection).

**Inference efficiency.** Generally, numerical weather prediction models (e.g., IFS) require around 50 minutes for the medium-range global forecast, while deterministic ML weather prediction models take less than 1 minute (Rasp et al., 2020) but cannot model the weather uncertainty. CoDiCast needs about 12 minutes (see the last row in Table 3) for the global weather forecast, potentially balancing the efficiency and accuracy with essential uncertainty quantification. The efficiency also depends on the model complexity.

Table 2: Continuous ranked probability scores (CRPS) (↓) with 24 hours lead time IFS ENS are from (Rasp et al., 2024).

| Model | Z500 | T850 | T2m | U10 | V10 |
|---|---|---|---|---|---|
| CoDiCast | 86.03 | 0.63 | 0.61 | 0.71 | 0.76 |
| IFS ENS | 24.76 | 0.36 | 0.37 | 0.55 | 0.56 |

### 5.4 QUALITATIVE EVALUATION

In Figure 6, we qualitatively evaluate the performance of CoDiCast on global forecasting tasks for all target variables, Z500, T850, T2m, U10 and V10 at the lead time of 6 hours. The first row is the ground truth of the target variable, the second row is the prediction and the last row is the difference between the model prediction and the ground truth. From the scale of their color bars, we can tell that the error percentage is less than 3% for variables Z500, T850, and T2m. Nevertheless, error percentages over 50% exist for U10 and V10 even though only a few of them exist. Furthermore, we observe that most higher errors appear in the high-latitude ocean areas, probably due to the sparse data nearby. We provide visualizations for longer lead times (up to 3 days) in Appendix F.

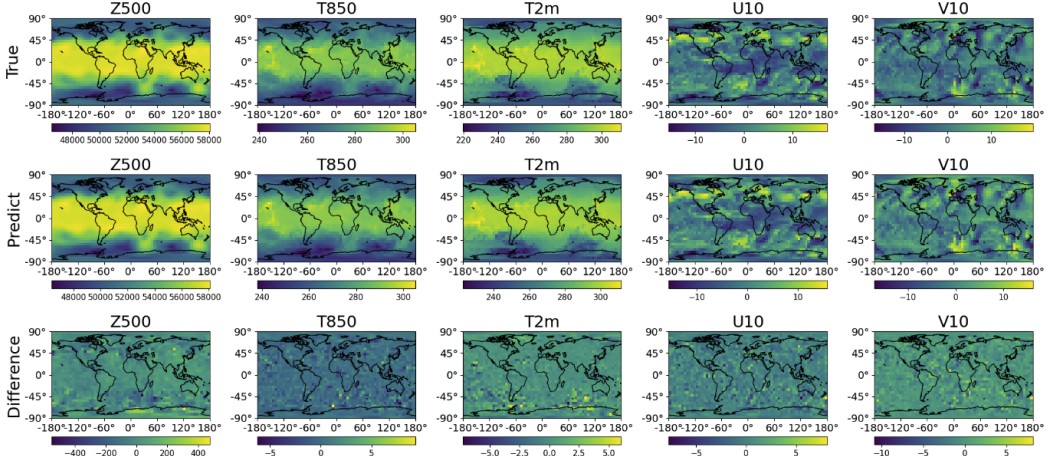

Figure 6: Visualization of true and predicted values at 6 hours lead time.

### 5.5 ABLATION STUDY

CoDiCast includes two significant components: *pre-trained encoder* and *cross attention*. To study their effectiveness, we conduct an ablation study as follows: (a) **No-encoder** directly considers past observations as conditions to diffusion model; (b) **No-cross-attention** simply concatenate the embedding and the noisy sample at each denoising step; (c) **No-encoder-cross-attention** concatenate the past observations and the noisy sample at each denoising step. From the results in Figure 7, we can observe that the full version of CoDiCast consistently outperforms all other variants, demonstrating both components positively contribute to generating plausible weather scenarios.

### 5.6 PARAMETER STUDY

**Diffusion step.** We try various diffusion steps $N = \{250, 500, 750, 1000, 1500, 2000\}$. Table 3 shows that the accuracy improves as the number of diffusion steps increases when $N < 1000$, indicating that more intermediate steps are more effective in learning the imperceptible attributes during the denoising process. However, when $1000 < N < 2000$, the accuracy remains approximately flat but the inference time keeps increasing linearly. Considering the trade-off between accuracy and efficiency, we finally set $N = 1000$ for all experiments.

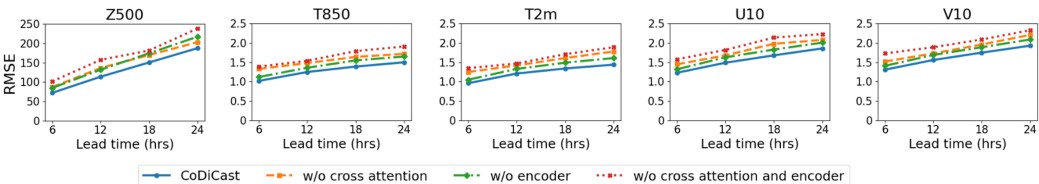

Figure 7: Ablation study to study the effect of pre-trained `encoder` and `cross-attention`.

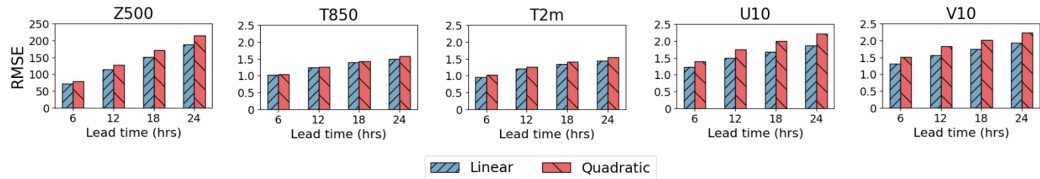

Figure 8: Effect of linear and quadratic variance scheduling methods.

**Method for variance scheduling.** We use the same start and end variance value, $\beta$, as DDPM (Ho et al., 2020) where $\beta \in [0.0001, 0.02]$. We study the effect of "linear" and "quadratic" variance scheduling in this section. The results are provided in Figure 8. It shows that the "linear" variance scheduling provides better performance than "quadratic" one for variables `Z500`, `T2m`, `U10`, and `V10`, while the performance of both "linear" and "quadratic" modes is roughly same for variable `T850`. Therefore, "linear" variance scheduling is utilized in our `CoDiCast` model.

Table 3: Latitude-weighted RMSE with various diffusion steps. We mark the lowest scores in bold font. The last row represents the inference time of `CoDiCast`.

| Variable | Lead | Diffusion Step | | | | | |
|---|---|---|---|---|---|---|---|
| | | 250 | 500 | 750 | 1000 | 1500 | 2000 |
| Z500 | 6 | 341.1 | 187.9 | 121.2 | **73.1** | 73.7 | 75.3 |
| | 12 | 359.6 | 178.7 | 116.9 | **114.2** | 117.2 | 118.6 |
| | 18 | 664.6 | 331.6 | 189.2 | **152.4** | 155.7 | 156.2 |
| | 24 | 696.1 | 324.8 | 190.6 | **186.5** | 193.5 | 191.9 |
| | 36 | 973.8 | 472.6 | **255.9** | 256.8 | 267.3 | 262.7 |
| T850 | 6 | 2.41 | 1.65 | 1.31 | **1.02** | 1.04 | 1.05 |
| | 12 | 2.33 | 1.65 | 1.27 | **1.26** | 1.28 | 1.31 |
| | 18 | 3.94 | 2.25 | 1.47 | **1.41** | 1.43 | 1.45 |
| | 24 | 3.88 | 2.38 | 1.53 | **1.52** | 1.56 | 1.58 |
| | 36 | 5.32 | 3.14 | 1.82 | **1.75** | 1.79 | 1.81 |
| T2m | 6 | 3.06 | 1.75 | 1.29 | **0.95** | 0.98 | 0.99 |
| | 12 | 3.25 | 1.73 | 1.26 | **1.21** | 1.26 | 1.27 |
| | 18 | 5.41 | 2.62 | 1.58 | **1.34** | 1.39 | 1.42 |
| | 24 | 5.26 | 2.79 | 1.63 | **1.44** | 1.50 | 1.53 |
| | 36 | 7.07 | 3.74 | 1.97 | **1.65** | 1.70 | 1.78 |
| U10 | 6 | 1.90 | 1.62 | 1.49 | **1.24** | 1.31 | 1.35 |
| | 12 | 1.92 | 1.59 | **1.42** | 1.50 | 1.60 | 1.64 |
| | 18 | 2.65 | 2.04 | 1.77 | **1.68** | 1.79 | 1.83 |
| | 24 | 2.74 | 2.05 | **1.81** | 1.87 | 1.99 | 2.01 |
| | 36 | 3.65 | 2.64 | 2.19 | **2.25** | 2.36 | 2.40 |
| V10 | 6 | 1.87 | 1.63 | 1.54 | **1.30** | 1.37 | 1.41 |
| | 12 | 1.79 | 1.64 | 1.56 | **1.56** | 1.67 | 1.69 |
| | 18 | 2.47 | 2.01 | 1.84 | **1.75** | 1.85 | 1.88 |
| | 24 | 2.43 | 2.11 | **1.89** | 1.94 | 2.04 | 2.06 |
| | 36 | 3.21 | 2.55 | **2.18** | 2.35 | 2.46 | 2.47 |
| Inference time (min) | | $\sim 3$ | $\sim 6$ | $\sim 10$ | $\sim 12$ | $\sim 20$ | $\sim 27$ |

## 6 CONCLUSIONS

In this work, we start with analyzing the limitations of current deterministic numerical weather prediction (NWP) and machine-learning weather prediction (MLWP) approaches—they either cause substantial computational cost or lack uncertainty quantification in the forecasts. To address these limitations, we propose a conditional diffusion model, `CoDiCast`, which contains a conditional *pre-trained encoder* and a *cross-attention* component. Quantitative and qualitative experimental results demonstrate it can simultaneously complete more accurate predictions than existing MLWP-based models and a faster inference than NWP-based models while being capable of providing uncertainty quantification compared to deterministic methods. In conclusion, `CoDiCast` achieves a critical trade-off between high accuracy, high efficiency, and low uncertainty for global weather prediction.

**Limitation and Future work.** We use low-resolution (5.625°) data currently due to the relatively slow inference process of diffusion models compared to deterministic ML models. In the future, we will focus on accelerating diffusion models (Song et al., 2020) to adapt the higher-resolution data. Besides the meteorological numerical data, weather events are often recorded or reported in the form of text. We will study how to leverage LLMs to extract their implicit interactions (Li et al., 2024) and inject them into diffusion models to guide the generation process.

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

APPENDIX

## A  DATASET

We introduce a detailed description of the ERA5 dataset. As the predominant data source for learning and benchmarking weather prediction systems, the ERA5 reanalysis archive from the European Center for Medium-Range Weather Forecasting (ECMWF) provides reanalyzed data from 1979 onwards. This data is available on a $0.25° \times 0.25°$ global latitude-longitude grid of the Earth's sphere, at hourly intervals, with different atmospheric variables at 37 different altitude levels and some variables on the Earth's surface. The grid overall contains $721 \times 1440$ grid points for latitude and longitude, respectively. Due to the limited computational resources, we used the preprocessed version of ERA5 from WeatherBench Rasp et al. (2020) in our work. This dataset[2] contains re-gridded ERA5 reanalysis data in three lower resolutions: $5.625°$, $2.8125°$, and $1.40625°$. To guarantee fair comparison with the benchmarks Verma et al. (2024), we follow the `ClimODE` work and choose the $5.625°$ resolution dataset for variables: geopotential at 500 hPa pressure level (`Z500`), atmospheric temperature at 850 hPa pressure level (`T850`), ground temperature (`T2m`), 10 meter U wind component (`U10`) and 10 meter V wind component (`V10`). *Single* represents surface-level variables, and *Atmospheric* represents time-varying atmospheric properties at chosen altitudes. A sample at a certain time point can be represented by $X^t \in \mathbb{R}^{H \times W \times C}$ where $H \times W$ refers to the spatial resolution of data which depends on how densely we grid the globe over latitude and longitude, $C$ refers to the number of channels (i.e, weather variables). In our work, $H, W$, and $C$ are $32, 64$, and $5$, accordingly. Notably, both `Z500` and `T850` are two popular verification variables for global weather prediction models, while `T2m`, `U10`, and `V10` directly pertain to human activities.

Table 4: Variable Information.

| Type | Variable | Abbrev. | ECMWF ID | Levels | Range | Unit |
|------|----------|---------|----------|--------|-------|------|
| Single | 2 metre temperature | T2m | 167 | | $[193.1, 323.6]$ | $K$ |
| Single | 10 metre U wind | U10 | 165 | | $[-37.3, 30.2]$ | $m/s$ |
| Single | 10 metre V wind | V10 | 166 | | $[-31.5, 32.5]$ | $m/s$ |
| Atmospheric | Geopotential | Z | 129 | 500 | $[43403.6, 59196.9]$ | $m^2/s^2$ |
| Atmospheric | Temperature | T | 130 | 850 | $[217.9, 313.3]$ | $K$ |

## B  MODEL ARCHITECTURE

We present the detailed architectures of the autoencoder, cross-attention block, and U-Net model used in our work. Meanwhile, we also illustrate how we organize the input data and how they flow through different machine-learning model blocks. We recommend readers check out Figures 2, 3, and 4 while looking into the following architectures.

### B.1  AUTOENCODER

We train an autoencoder model consisting of two main parts: an encoder and a decoder. The encoder compresses the input to feature representation (embedding) in the latent space. The decoder reconstructs the input from the latent space. After training, the pre-trained encoder can be extracted to generate embedding for input data. In our work, the convolutional autoencoder architecture is designed for processing spatiotemporal weather data at a time point, $t$, represented as $X^t \in \mathbb{R}^{H \times W \times C}$. The encoder consists of a series of convolutional layers with $2 \times 2$ filters, each followed by a `ReLU` activation function. The layers have 32, 128, 256, and 512 filters, respectively, allowing for a progressive increase in feature depth, thereby capturing essential patterns in the data. The decoder starts with 512 filters and reduces the feature depth through layers with 256 and 128 filters, each followed by `ReLU` activations. This design ensures the reconstruction of the input data while preserving the learned features, enabling the model to extract meaningful embeddings that encapsulate the spatiotemporal characteristics of the input.

---

[2]`https://github.com/pangeo-data/WeatherBench`

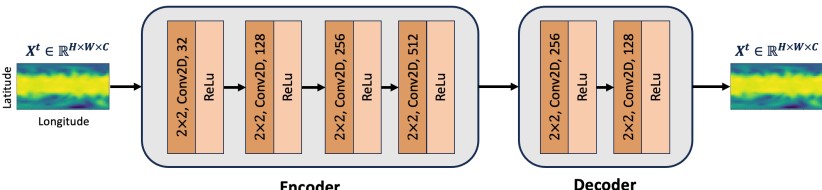

Figure 9: Architecture of the Autoencoder model.

## B.2 CROSS-ATTENTION

The cross-attention is used to learn the interaction between past observations and the noisy data at each diffusion step. We consider the past observations as the conditions to guide the diffusion models during generation. Given the weather states in the past two time points, $X^{T \times H \times W \times C}$, we utilize the pre-trained encoder to learn the embedding from each time point, $X^{T \times H \times W \times d_e}$. To better use the *attention* mechanism, we first reshape it to $X^{(H*W) \times (T*d_e)}$ and convert it to key and value matrices: $K \in \mathbb{R}^{(H*W) \times d_k}$ and $V \in \mathbb{R}^{(H*W) \times d_v}$. We consider the noisy sample at each diffusion step, $X_n \in \mathbb{R}^{T \times H \times W \times C}$, as a query. It is transformed to $Q \in \mathbb{R}^{(H*W) \times d_q}$. Then, the *cross-attention* mechanism is implemented by $Attention(Q, K, V) = softmax(\frac{QK^T}{\sqrt{d}}) \cdot V$. In our work, we set $d_q = d_k = d_v = d = 64$ where $d$ is the projection embedding length.

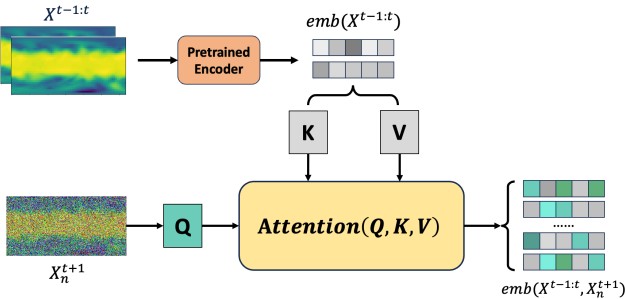

Figure 10: Architecture of the cross-attention block.

## B.3 U-NET

Our U-Net architecture is similar to that[3] of DDPM Ho et al. (2020) but with necessary changes to adapt to the problems in this work. Each U-Net unit comprises two ResNet blocks He et al. (2016) and a convolutional up/downsampling block. Self-attention was included between the convolution blocks once we reached a specific resolution ($4 \times 8, 2 \times 4$), represented in blue arrows. We employ four U-Net units for both the downsampling and upsampling processes. We use *MaxPooling* in the downsampling units where the channel dimension is $64 \times j$ ($j = \{1, 2, 3, 4\}$ refers to the layer index). The upsampling units follow the reverse order. We set the upsampling factor as 2 and the "nearest" interpolation. We used the `swish` activation function throughout the network. We also had `GroupNormalization` layer for more stable training where the number of groups for Group Normalization is 8. Group Normalization divides the channels into groups and computes within each group the mean and variance for normalization.

Notably, for the target variable, $X^{t+1}$ at the $n$ diffusion step, the input to U-Net involves the mixture embedding of past weather states, $X^{t-1:t}$, and the noisy sample from the last diffusion step, $X_n^{t+1}$. The mixture embedding is obtained by the cross-attention mechanism described above. The channel dimension output is five because of five weather variables of interest to predict. This is achieved by a convolutional layer with a $1 \times 1$ kernel.

---

[3]`https://github.com/hojonathanho/diffusion/blob/master/diffusion_tf/`
`models/unet.py`

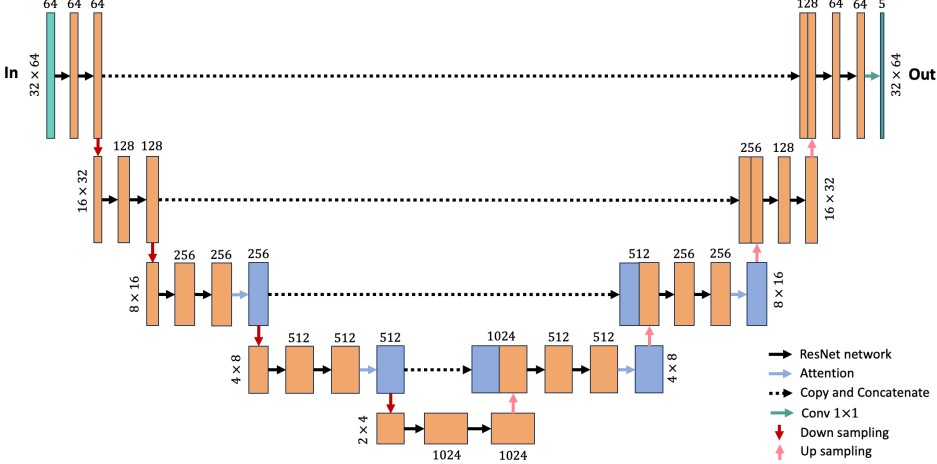

Figure 11: Architecture of the U-Net model.

## C  TRAINING DETAILS

We provide the hyperparameters for training our model `CoDiCast`, which includes pre-training `Autoencoder` and training the `denoiser` network. Since it is more helpful to find the minimum loss if using a decayed learning rate as the training progresses, we applied an exponential decay function to an optimizer step given a provided initial learning rate.

Table 5: Hyperparameters of Training Autoencoder.

| Abbreviation | Training Autoencoder | Training Denoiser |
|---|---|---|
| Epochs | 100 | 800 |
| Batch_size | 128 | 256 |
| Learning_rate | 1e-4 | 2e-4 |
| Decay_steps | 10000 | 10000 |
| Decay_rate | 0.95 | 0.95 |

## D  EVALUATION METRICS

**Root Mean Square Error.**  Following (Verma et al., 2024), we assess the model performance using latitude-weighted Root Mean Square Error (RMSE). RMSE measures the average difference between values predicted by a model and the actual values.

$$\text{RMSE} = \frac{1}{M} \sum_{m=1}^{M} \sqrt{\frac{1}{H \times W} \sum_{h=1}^{H} \sum_{w=1}^{W} L(h)(\tilde{X}_{m,h,w} - X_{m,h,w})^2},$$

where $L(h) = \frac{1}{H} cos(h) \sum_{h'}^{H} cos(h')$ is the latitude weight and $M$ represents the number of test samples.

**Anomaly Correlation Coefficient.**  ACC is the correlation between prediction anomalies $\tilde{X}'$ relative to climatology and ground truth anomalies $\hat{X}$ relative to climatology. ACC is a critical metric in climate science to evaluate the model's performance in capturing unusual weather or climate events.

$$\text{ACC} = \frac{\sum_{m,h,w} L(h) \tilde{X}'_{m,h,w} X'_{m,h,w}}{\sqrt{\sum_{m,h,w} L(h) \tilde{X}'^2_{m,h,w} \cdot \sum_{m,h,w} L(h) X'^2_{m,h,w}}},$$

where observed and forecasted anomalies $X' = X - C, \tilde{X}' = \tilde{X} - C$, and climatology $C = \frac{1}{M} \sum_m X$ is the temporal mean of the ground truth data over the entire test set.

**Continuous Ranked Probability Score.** Following (Rasp et al., 2024) we utilize the continuous ranked probability score (CRPS) as a probabilistic metric to measure the discrepancy between the predicted distribution and a single ground-truth value. It is a generalization of the MAE for distributional predictions. CRPS penalizes over-confidence in addition to inaccuracy in ensemble predictions—a lower CRPS is better. More specifically, it is a score function that compares the ground truth target $y$ with the cumulative distribution function (CDF) $F$ of the prediction:

$$CRPS(D, y) = \int (F_D(x) - \mathbb{1}_{\{x \geq y\}})^2 \, dx,$$

where $F_D$ is the cumulative distribution function of the forecasted distribution $D$, $\mathbb{1}$ is the indicator function (or Heaviside step function), and $y \in \mathbb{R}$ is the scalar observation. Based on the work (Gneiting & Raftery, 2007), the continuous ranked probability score can be written as:

$$CRPS(D, y) = \mathbb{E}_{X \sim D}[|X - y|] - \frac{1}{2}\mathbb{E}_{X, X' \sim D}[|X - X'|]$$

where $X$ and $X'$ are independent and identically distributed (*iid*) samples from the distributional prediction $D$. We use the non-parametric "fair estimate to the CRPS" (Ferro, 2014) estimating $D$ with the empirical CDF of $n = 20$ *iid* samples $X_i \sim D$:

$$\hat{CRPS}(X, y) = \frac{1}{n} \sum_{i=1}^{n} |X_i - y| - \frac{1}{2n(n-1)} \sum_{i=1}^{n} \sum_{j=1}^{n} |X_i - X_j|,$$

where the first term is the MAE between the target and samples of the predictive distribution, while the second term is small for small predictive variances, vanishing completely for point estimates.

# E  EXPERIMENTAL RESULTS WITH LONGER LEAD TIMES

In this section, we compare `CoDiCast` against the other two baselines for the longer lead time. We observe that `CoDiCast` shows more accurate and robust performance on ACC scores. Additionally, it still performs the best for the 3-day forecast on RMSE scores, but its performance gradually drops as the lead time increases up to 6 days.

Table 6: Latitude-weighted RMSE ($\downarrow$) and ACC ($\uparrow$) comparison with baselines on global weather forecasting. We mark the scores in bold if `CoDiCast` performs the best.

| Variable | Lead Time | RMSE ($\downarrow$) | | | ACC ($\uparrow$) | | |
|---|---|---|---|---|---|---|---|
| | | ClimaX | ClimODE | CoDiCast | ClimaX | ClimODE | CoDiCast |
| Z500 | 72 | 687.0 | 478.7$\pm$48.3 | **451.6**$\pm$39.5 | 0.73 | 0.88 | **0.92** |
| | 144 | 801.9 | 783.6$\pm$37.3 | 825.5$\pm$45.2 | 0.58 | 0.61 | **0.78** |
| T850 | 72 | 3.17 | 2.58$\pm$0.16 | **2.54**$\pm$0.14 | 0.76 | 0.85 | **0.93** |
| | 144 | 3.97 | 3.62$\pm$0.21 | 3.81$\pm$0.19 | 0.69 | 0.77 | **0.85** |
| T2m | 72 | 2.87 | 2.75$\pm$0.49 | **2.39**$\pm$0.37 | 0.83 | 0.85 | **0.96** |
| | 144 | 3.38 | 3.30$\pm$0.23 | 3.45$\pm$0.22 | 0.83 | 0.79 | **0.91** |
| U10 | 72 | 3.70 | 3.19$\pm$0.18 | **3.15**$\pm$0.19 | 0.45 | 0.66 | **0.71** |
| | 144 | 4.24 | 4.02$\pm$0.12 | 4.45$\pm$0.15 | 0.30 | 0.35 | **0.42** |
| V10 | 72 | 3.80 | 3.30$\pm$0.22 | **3.26**$\pm$0.14 | 0.39 | 0.63 | **0.68** |
| | 144 | 4.42 | 4.24$\pm$0.10 | 4.51$\pm$0.17 | 0.25 | 0.32 | **0.37** |

# F  VISUALIZING PREDICTION WITH LONGER LEAD TIMES

We provide the forecast at longer lead times (i.e., $24, 36, 72$ hours). The first row is the ground truth of the target variable, the second row is the prediction of `CoDiCast` and the last row is the difference between the model prediction and the ground truth.

## F.1 SHORT RANGE WEATHER FORECASTING

Short-range weather forecasting at the 24-hour lead time for all target variables.

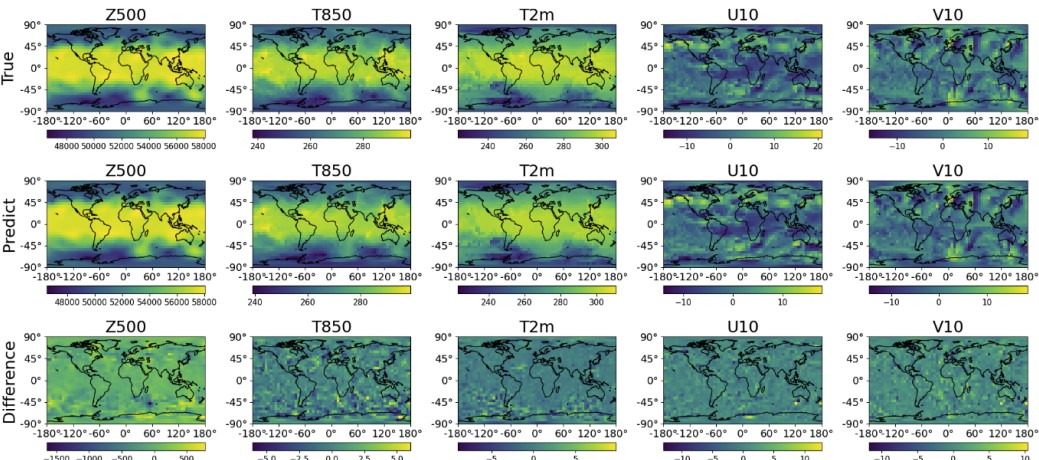

Figure 12: Visualizations of true and predicted values of all five variables at 24 hours lead time.

## F.2 MEDIUM-RANGE WEATHER FORECASTING

Medium-range weather forecasting at the 36-hour lead time for all target variables.

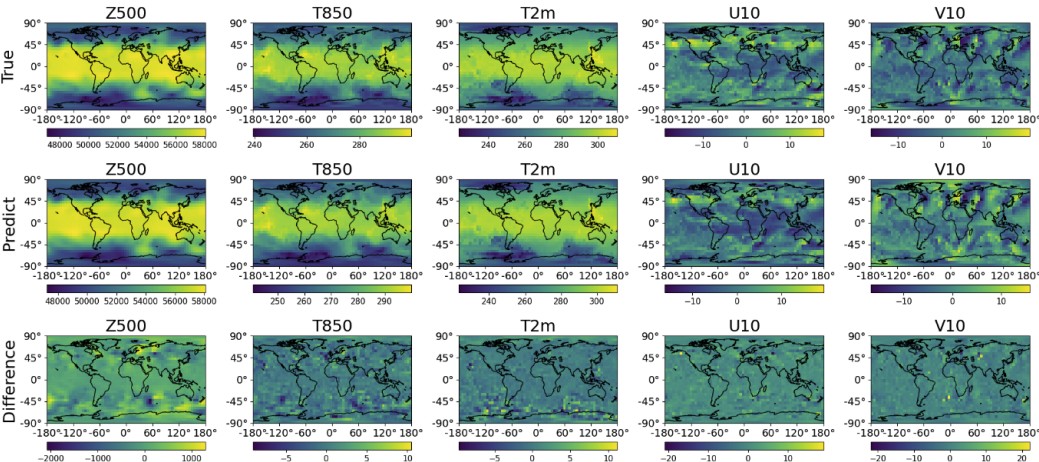

Figure 13: Visualizations of true and predicted values of all five variables at 36 hours lead time.

## F.3 LONG-RANGE WEATHER FORECASTING

Longer-range weather forecasting at the 72-hour lead time for all target variables.

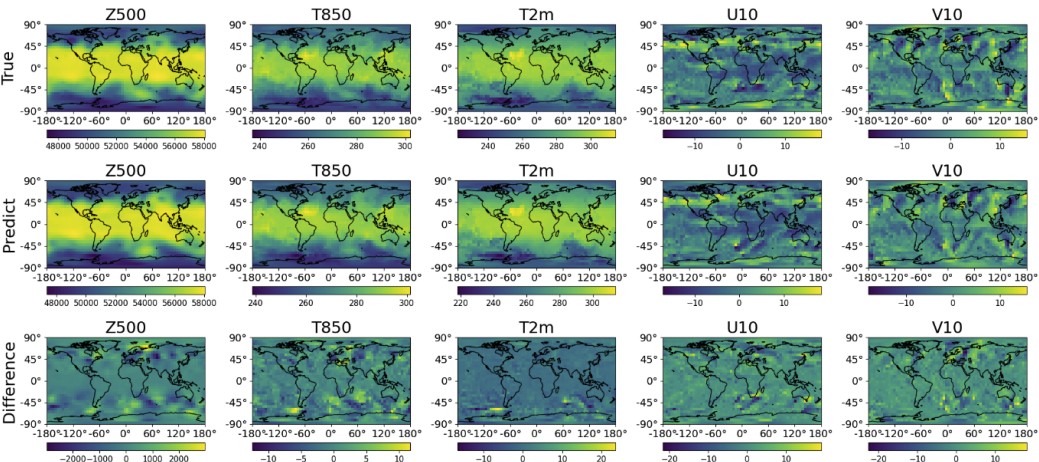

Figure 14: Visualizations of true and predicted values of all five variables at 72 hours lead time.

