# OpenReview forum: "CoDiCast: Conditional Diffusion Model for Weather Prediction with Uncertainty Quantification"
_ICLR.cc/2025/Conference — Submitted to ICLR 2025_

### Official Review · Reviewer_7BiF · 2024-10-27

**Soundness:** 2
**Presentation:** 3
**Contribution:** 1
**Rating:** 3
**Confidence:** 4

**Summary:**

The paper proposes CoDiCast, an autoregressive, diffusion-based weather model that generates ensemble forecasts. Using an additional embedding and attention-based UNet as the denoiser, it is able to produce reasonable forecast at 3-day (72-hour) lead-time.

**Strengths:**

The paper is clear and well-written. The motivation is also clearly laid out e.g., the need for ensemble forecasting to account for uncertainties of chaotic systems such as the weather dynamics. The use of additional embedding z as conditioning is also interesting.

**Weaknesses:**

The paper has several concerning weaknesses, most notably on novelty, over-simplified problem formulation, and the use of weak/inappropriate baselines, described in more details below:

__On novelty:__

1. The use of attention-based UNet is not novel and has been used extensively in prior work e.g., [1]. Even cross-attention has been used extensively in similar task e.g., ClimaX.

2. The application of diffusion-based approach for weather modeling is likewise not novel, as is rightly pointed out by the authors in GenCast and other similar works that come after that, e.g., SEEDS [2] where the code is open-sourced. Although the claim about GenCast not being open-sourced is valid, the application of CoDiCast on much coarser spatial resolution (e.g., 5.625-degree vs the typical 0.25-degree in weather forecasting; ~400 smaller horizontal domain), limited temporal extent (3-day vs 10-day), and small number of variable (5 vs 100+ variables) is unconvincing, given that the inference time can go up to 12-27 minutes (which is much, much worse even for IFS ENS if all spatiotemporal resolution / number of resolved variables are made identical).

__On inappropriate/weak baselines:__

3. The baselines used are mostly deterministic (FourCastNet, ClimaX), and are not a fair baseline for a probabilistic modeling framework. The authors could use open-sourced diffusion-based weather model as baseline e.g., SEEDS [2]. Otherwise, the authors can consider a probabilistic formulation of such deterministic model through e.g., MC-dropout or IC perturbation approaches [3].

4. Furthermore, the inferior performance of CoDiCast even for the control (deterministic) run of IFS (Table 1) across lead-time is concerning since many SOTAs e.g., GraphCast have outperformed IFS control runs at such short lead time of 3 days.

In all, the lack of novelty, overly-simplified problem setting, and weak/inappropriate baselines are serious issues that motivate my reject rating.


__References:__

[1] Oktay, Ozan, et al. "Attention u-net: Learning where to look for the pancreas." arXiv preprint arXiv:1804.03999 (2018).

[2] Li, Lizao, et al. "Generative emulation of weather forecast ensembles with diffusion models." Science Advances 10.13 (2024): eadk4489.

[3] Rühling Cachay, Salva, et al. "Dyffusion: A dynamics-informed diffusion model for spatiotemporal forecasting." Advances in Neural Information Processing Systems 36 (2024).

**Questions:**

1. What is the number of ensemble members used to generate the ensemble forecasts?
2. What is the number of parameters used in CoDiCast, including the autoencoder and denoiser module.

---

### Official Review · Reviewer_YGpR · 2024-11-01

**Soundness:** 2
**Presentation:** 3
**Contribution:** 2
**Rating:** 3
**Confidence:** 5

**Summary:**

The authors identify that a drawback of many existing ML weather forecasting models is that they only give deterministic forecasts, and do not capture uncertainty. To remedy this a diffusion model is proposed, capable of generating ensemble forecasts. The proposed CoDiCast model builds on the DDPM framework and uses a U-net as denoiser. Conditioning on the weather states at previous time points is achieved with a cross-attention mechanism, attending to representations learned through pre-training an autoencoder. Experiments are conducted on 5.625 degree ERA5 data. CoDiCast achieves lower errors than comparable deterministic baselines on forecasting up to 36 h.

**Strengths:**

1. The work is contributing to pushing the research front of ML weather prediction. Creating probabilistic forecasting models capable of creating ensemble forecasts is an important and relevant step for ML weather prediction. This paper is a contribution in this direction.
2. The way conditioning information is incorporated in the denoising network is novel to weather forecasting. Learning latent representations of previous states through an autoencoder and then utilizing cross-attention seems like a reasonable way to make the model utilize the conditioning information.
3. The paper is overall well-written, and captures many of the key details that are important for understanding how the diffusion model framework is used for generating ensemble forecasts.
4. The proposed model shows good performance on short-term weather forecasting when comparing to baselines trained on the same data.
5. Multiple interesting ablation studies are given in section 5.6. These show that all modeling components for incorporating the conditioning information contribute to the model performance. This section also contains an insightful study on the impact of the number of diffusion steps used for sampling.

**Weaknesses:**

**Major**
1. The paper lacks contextualization within ML weather forecasting literature dealing with uncertainty. There are multiple works apart from Price et al. (2024) that develop methods for ML ensemble forecasting. I am missing:
    * A proper discussion about these different methods for ML ensemble weather forecasting in the related work section. These include perturbation-based methods e.g. (Chen et al., 2023), (Bülte et al., 2024) and multi-model ensembles (Weyn et al., 2021). In particular, other generative ML weather models based on latent variable formulations have been proposed (Hu et al., 2023), (Oskarsson et al., 2024) and it seems necessary to discuss how the proposed method relates to these, given its high sampling cost.
    * Some comparison of the uncertainty estimation capabilities of the model to any of the methods discussed above. Currently the only comparison related to uncertainty is to the IFS ensemble. While this is interesting, given the body of work on uncertainty and ensemble forecasting for ML weather models in the literature some comparison to these methods can be expected.
    * A discussion of, and comparison to, the uncertainty estimates of ClimODE. ClimODE is one of the considered baselines and much of the experiment setup follows closely Verma et al. (2024). While not an ensemble forecasting model, ClimODE provides uncertainty estimates. There is no clear reason why these are not considered, for example when computing CRPS in Table 2.
2. The main usefulness of ensemble weather forecasting is for lead times in the medium-  to long-range (up to 10-15 days). At these time points there is great uncertainty in the atmosphere and the chaotic nature of the atmosphere can give rise to highly complex and multi-modal distributions. At shorter lead times (up to a couple days) there is far less uncertainty, especially when considering coarse resolutions. Still the authors choose to focus on these shorter lead times, demonstrating results in the main paper up to 24 or 36 h. While uncertainty quantification is valuable also at these lead times, it is not the main promise of ensemble weather forecasting. More worrying is the fact that when considering longer lead times (Table 6 in the appendix) the proposed method shows worse performance than baselines.
3. Constructing a model providing uncertainty estimates is the central contribution of the paper. Still, the evaluation of the uncertainty estimates from the model is not convincing. The only quantitative evaluation of uncertainties is CRPS in table 2, which lacks any comparison to ML baselines. Considering only CRPS does also not say much about how calibrated these uncertainties are, indicating to what extent they can be trusted. For ensemble forecasting this is typically measured by using spread-skill-ratio or rank histograms (see e.g. evaluation in Price et al. (2023)), which would be good to see also here.
4. Sampling from the model seems prohibitively slow. The authors write that sampling a forecast (I interpret this as one ensemble member, please clarify otherwise) takes 12 minutes. This is for 5 variables at 5.625 degree resolution. This can be compared to the GenCast diffusion model (Price et al., (2023)) that samples forecasts of 80 variables at 0.25 degree resolution in 8 minutes. This slowness is likely caused by having to take 1000 sampling steps, which can be compared to the 20 in GenCast. The authors show however that using fewer sampling steps in CoDiCast leads to drastically worse results. Moreover, when considering other methods for creating ML weather model ensembles (see references under point 1) that produce forecasts in seconds, this seems like an even bigger drawback. The authors point out that the forecast time is still faster than IFS, but this seems of limited relevance as this is the case for all ML models and IFS models many more variables at a much higher resolution.

**Minor**
1. The ICLR audience can at this point be expected to have a basic idea of how a diffusion model works, sufficiently to understand its use in this paper with some references given. Background section 3.2 could thus be significantly shortened or even moved to appendix. Sections 4.1 and 4.2 also describe straightforward instantiation of the DDPM framework with conditioning on $\tilde{Z}$, and seem superfluous. The description of the DDPM framework is however nicely written and pedagogical, so could be nice to keep in appendix and refer unfamiliar readers to.
2. Using a standard U-net does not respect any of the geometry or periodicity of earth coordinates. The paper does not describe any changes to the network to handle periodic boundary conditions in the longitudinal direction.
3. In Figure 6 sampled forecasts from the model are visualized for a lead time of 6 h. At the considered resolution there is not a large difference between the initial state and the state after 6 h for most variables. This is thus not a very interesting figure and showing predictions for later lead times in the main paper would be far more interesting.

*References:*

* Price, Ilan, et al. "Gencast: Diffusion-based ensemble forecasting for medium-range weather." arXiv preprint arXiv:2312.15796 (2023).
* Chen, Lei, et al. "FuXi: A cascade machine learning forecasting system for 15-day global weather forecast." npj Climate and Atmospheric Science 6.1 (2023): 190.
* Bülte, Christopher, et al. "Uncertainty quantification for data-driven weather models." arXiv preprint arXiv:2403.13458 (2024).
* Weyn, Jonathan A., et al. "Sub‐seasonal forecasting with a large ensemble of deep‐learning weather prediction models." Journal of Advances in Modeling Earth Systems 13.7 (2021): e2021MS002502.
* Hu, Yuan, et al. "SwinVRNN: A Data‐Driven Ensemble Forecasting Model via Learned Distribution Perturbation." Journal of Advances in Modeling Earth Systems 15.2 (2023): e2022MS003211.
* Oskarsson, Joel, et al. "Probabilistic Weather Forecasting with Hierarchical Graph Neural Networks." arXiv preprint arXiv:2406.04759 (2024).
* Verma, Yogesh, Markus Heinonen, and Vikas Garg. "ClimODE: Climate and Weather Forecasting with Physics-informed Neural ODEs." The Twelfth International Conference on Learning Representations (2024).

**Questions:**

1. When computing RMSE, is this done for the ensemble mean or just a single sample from the model? I wonder this for all occurrences of RMSE metrics in the paper. If the ensemble mean is used, how many ensemble members were sampled? Given that a probabilistic model is constructed, that describes a distribution over future states, each forecast is a sample. The RMSE as a metric is minimized by the mean of the distribution, and one would not expect single samples to achieve low RMSE (this is desirable, as if the model accurately captures uncertainties the samples can be far from the mean). The RMSE of interest in any comparison would rather be that of the ensemble mean. This methodological choice is crucial for how the results in the paper should be judged, and important to clarify.
2. What is the total dimensionality of $\tilde{Z}^t$? From the descriptions given of the autoencoder I computed this to be $2\times4\times512=4096$, and input $X^t$ having dimensions $64 \times 32 \times 5 = 10240$. If that is correct that does not seem like a substantial reduction in dimensionality. Could you comment on this choice of dimensionality for $\tilde{Z}^t$?
3. What do the error ranges in gray in Table 1 represent? Is this variation across multiple models re-trained with different random seeds?
4. Is the CRPS values in Table 2 computed for the full test set or only for the case study scenario showed in Figure 5?

---

### Official Review · Reviewer_iYt9 · 2024-11-03

**Soundness:** 3
**Presentation:** 3
**Contribution:** 1
**Rating:** 3
**Confidence:** 4

**Summary:**

The paper proposes CoDiCast, a conditional diffusion model for probabilistic global weather forecasting. Similar to other related methods, CoDiCast is 2nd order autoregressive (i.e. conditions on two previous state variables to forecast/generate the next weather state) and is is trained on ERA5. A key feature of CoDiCast is that the conditioning states are encoded using a pretrained autoencoder before being fed as inputs to the denoising network in the diffusion model.

**Strengths:**

The paper tackles an important problem and the proposed solution is shown to be an efficient approach to probabilistic weather forecasting. The use of a pretrained autoencoder to encode the conditioning states has, to the best of my knowledge, not been used in this application domain before and is shown to result in a performance boost.

**Weaknesses:**

I don’t believe that this paper has sufficient novelty. Diffusion models have been used for probabilistic weather forecasting (in very similar way to the current paper) in the GenCast model (https://arxiv.org/abs/2312.15796). I understand that the code for this method is not public, which is perhaps a valid argument for not comparing with it in a numerical evaluation, but it does not change the fact that a very similar method has been published before. Furthermore, diffusion models have been used for other spatio-temporal forecasting problems (see e.g. the survey https://arxiv.org/abs/2404.18886 or https://arxiv.org/abs/2309.01745 for autoregressive diffusion models similar to CoDiCast, evaluated on turbulent flow simulations).

**Questions:**

The main novelty in this paper, as far as I can tell, is the use of a pretrained autoencoder for encoding the conditioning states. If this is indeed a critical design choice which results in a consistent performance boost compared to alternative approaches, this could be a valuable contribution. However, the effect of this design choice is only evaluated in a short ablation study (Fig 7) and it is not clear how general the conclusions from this example are. For instance, how does the pretrained encoder compare with using a learnable encoder that is trained jointly with the denoiser? How does it compare with the approach in https://arxiv.org/pdf/2309.01745, based on jointly noising the conditioning and target states? How robust is the performance improvement to different architectures, noise schedules, and other hyperparameters?

If the authors can show consistent improvements using a pretrained autoencoder in this context, then I would recommend rewriting the paper to emphasize this aspect, and focus on a rigorous empirical evaluation to prove this point.

---

### Official Review · Reviewer_VcfD · 2024-11-04

**Soundness:** 2
**Presentation:** 3
**Contribution:** 2
**Rating:** 5
**Confidence:** 3

**Summary:**

In this paper, the authors propose CoDiCast, a conditional diffusion model for weather forecasting. One of the main motivations of using a probabilistic model is for uncertainty quantification.

**Strengths:**

- Important applied ML problem for weather forecasting where ML methods are needed and need more rigorous testing
- Uncertainty quantification in this application is also very needed
- High computational cost of ensembles with NWP methods is good motivation
- Good test case on global forecasting at small resolution of 5.625 degrees on the relevant ERA5 dataset
- Nice overview in the introduction on the the need for MLWP and relevant downstream tasks and need for UQ methods
- Nice overview on current MLWP models in the introduction (See Karlbauer et. al, "Comparing and Contrasting Deep Learning Weather Prediction Backbones on Navier-Stokes and Atmospheric Dynamics", 2024 https://arxiv.org/abs/2407.14129 for a benchmark comparison of these models)
- Good that the authors tested on ERA5 and used the CRPS metric to measure the performance of the uncertainty

**Weaknesses:**

- In the abstract, the authors state that existing methods have not shown high accuracy and low uncertainty. The high accuracy is good motivation but with uncertainty low is not guaranteed to be better so I would rephrase and clarify that.
- There are other methods to calculate uncertainty than ensembles (See Mouli et. al, "Using Uncertainty Quantification to Characterize and Improve Out-of-Domain Learning for PDEs", ICML, 2024 for a survey)
- There are also MLWP that calculate uncertainty and are not deterministic (See Gao et. al, "Prediff: Precipitation nowcasting with latent diffusion models", NeurIPS, 2023 and Price et. al, "GenCast: Diffusion-based ensemble forecasting for medium-range weather", 2024.
- I am concerned about the novelty since PreDiff also uses a conditional diffusion model.
- Adding MLWP benchmark paper  (Karlbauer et. al, "Comparing and Contrasting Deep Learning Weather Prediction Backbones on Navier-Stokes and Atmospheric Dynamics", 2024 https://arxiv.org/abs/2407.14129) for more detials on the pros/cons of current MLWP models
- 3 day prediction at 6 hour intervals is not very long and could be a limitation of the method if it cannot accurately forecast longer in time. How was this forecast horizon selected?
- In particular, this sentence "Moreover, a single deterministic NWP- and MLWP-based method cannot achieve uncertainty quantification." is too strong given the works of PreDiff and GenCast
- A lot of the background on DDPM in 3.2 can be moved to an appendix to save more space for the method and results
- PreDiff and GenCast should be added as baselines. It seems main contribution of this work is applying PreDiff which was tested on precipitation nowcasting problems to weather forecasting on ECMWF which I'm not sure is enough novelty for this conference.
- Can add FID score to the metrics

**Questions:**

1. How would the model apply on longer time-scales, especially climate lengthscales of 100 years out? I see some of this in Appendix E-F.
2. Please explain the clear methodology differences between the proposed methods and other probabilistic diffusion models for weather forecasting, e.g., PreDiff and GenCast?
3. PreDiff shows that using a Earthformer-UNet (Gao et., al, "Earthformer: Exploring Space-Time Transformers for Earth System Forecasting", NeurIPS, 2022) encoder, which is also an attention-based Transfomer model is better than the latent U-Net model in DDPM. Have the authors tested that here?
4. Why do the authors claim that smaller uncertainty is better? That is not necessarily the case and depends on whether it is well-calibrated with the error or not.

---

### Meta-Review · Area_Chair_uwz6 · 2024-12-29

**Metareview:**

Weather forecasting is a highly important task and ML has shown great promise towards this problem. This paper proposes a conditional diffusion model, where the conditioning leverages latent representations from an autoencoder, for weather prediction. The paper proposes to incorporate uncertainty quantification into the predictions by repeatedly sampling a Gaussian for each initial weather state and using that as input for the denoising process. Reviewers are rightfully concerned about limited novelty. One important shortcoming of the paper, as highlighted by one of the reviewers, is that uncertainty quantification isn't really critical for the prediction range in which the model is applied; uncertainty quantification becomes much more critical for long-range prediction and the method in the paper is falling short in those ranges.

**Additional Comments On Reviewer Discussion:**

The authors did not provide any rebuttal to address the reviewers' comments and questions about the paper.

---

### Decision · Program_Chairs · 2025-01-22

Reject